# Quantum transport of high-dimensional spatial information with a nonlinear detector

Bereneice Sephton[1], Adam Vallés [1,2,3] ✉, Isaac Nape [1], Mitchell A. Cox [4], Fabian Steinlechner [5,6], Thomas Konrad [7,8], Juan P. Torres [3,9], Filippus S. Roux[10] & Andrew Forbes [1] ✉

Information exchange between two distant parties, where information is shared without physically transporting it, is a crucial resource in future quantum networks. Doing so with high-dimensional states offers the promise of higher information capacity and improved resilience to noise, but progress to date has been limited. Here we demonstrate how a nonlinear parametric process allows for arbitrary high-dimensional state projections in the spatial degree of freedom, where a strong coherent field enhances the probability of the process. This allows us to experimentally realise quantum transport of high-dimensional spatial information facilitated by a quantum channel with a single entangled pair and a nonlinear spatial mode detector. Using sum frequency generation we upconvert one of the photons from an entangled pair resulting in high-dimensional spatial information transported to the other. We realise a $d = 15$ quantum channel for arbitrary photonic spatial modes which we demonstrate by faithfully transferring information encoded into orbital angular momentum, Hermite-Gaussian and arbitrary spatial mode superpositions, without requiring knowledge of the state to be sent. Our demonstration merges the nascent fields of nonlinear control of structured light with quantum processes, offering a new approach to harnessing high-dimensional quantum states, and may be extended to other degrees of freedom too.

Information exchange is the backbone of modern society, with our world connected by global networks of fibre and terrestrial links. Quantum technologies allow this exchange to be fundamentally secure, fuelling the nascent quantum global network[1]. For example, quantum key distribution exchanges a key from peer to peer (usually Alice and Bob) to decode the information transmitted between communicating parties[2], quantum secret sharing splits such a key amongst many nodes[3] and quantum secure direct communication sends it

without a key, but rather encoded in a transmitted quantum state[4]. In all these schemes, like its classical counterpart, the information is sent across a physical link between the sender and receiver. Remote state preparation[5,6] allows information exchange between parties without transmitting the information physically across the link, but the sender (Alice) must know the information to be sent. Teleportation[7–10] allows protected information exchange between distant parties without the need for a physical link[11], facilitated by the sharing of entangled

[1]School of Physics, University of the Witwatersrand, Wits, South Africa. [2]Molecular Chirality Research Center, Chiba University, Chiba, Japan. [3]ICFO - Institut de Ciencies Fotoniques, The Barcelona Institute of Science and Technology, Castelldefels, Barcelona, Spain. [4]School of Electrical and Information Engineering, University of the Witwatersrand, Johannesburg, South Africa. [5]Fraunhofer Institute for Applied Optics and Precision Engineering, Jena, Germany. [6]Friedrich Schiller University Jena, Abbe Center of Photonics, Jena, Germany. [7]School of Chemistry and Physics, University of KwaZulu-Natal, Durban, South Africa. [8]National Institute of Theoretical and Computational Sciences (NITheCS), KwaZulu-Natal, South Africa. [9]Department of Signal Theory and Communications, Universitat Politecnica de Catalunya, Barcelona, Spain. [10]National Metrology Institute of South Africa, Pretoria, South Africa. ✉e-mail: adam.valles@icfo.eu; andrew.forbes@wits.ac.za

photons and a classical communication channel, where the information sent must not be known by Alice.

All the aforementioned schemes would benefit from using high dimensional quantum states, offering higher channel capacity[12], security[13], or resilience to noise[14]. In the context of spatial modes of light as a basis, orbital angular momentum (OAM) has proven particularly useful and topical[15–17], as has path[18] and pixels[19], as potential routes towards high-dimensions. Yet experimental progress has been slow, with sharing keys shown up to $d = 6$ in optical fibre[20] and $d = 7$ in free-space[21], and sharing secrets up to $d = 11$[22]. Our interest is in schemes where the information is remotely shared and not physically sent, such as teleportation, which has been limited to $d = 2$ using OAM[23–25] and $d = 3$ using the path degree of freedom[26,27]. So far all of these approaches have used linear optics for their state control and detection, which has known limitations in the context of high-dimensional states[28]. More recently, nonlinear optics has emerged as an exciting creation, control and detection tool for spatially structured classical light[29], but has not found its way to controlling spatially structured quantum states beyond polarisation qubit measurement[30]. Although theoretical schemes have been proposed to use nonlinear approaches for high-dimensional quantum information processing and communication[31–33], none have yet been demonstrated experimentally.

Here, we experimentally demonstrate a nonlinear spatial quantum transport scheme for arbitrary dimensions using two entangled photons to form the quantum channel and a bright coherent source for information encoding. One of the photons from the entangled pair is upconverted in a nonlinear crystal using the coherent beam both as the information carrier and efficiency enhancer, with a successful single photon detection resulting in information transported to the other photon enabled by a bi-photon coincidence measurement. Our system works for spatial information in a manner that is dimension and basis independent, with the modal capacity of our quantum channel easily controlled by parameters such as beam size and crystal properties, which we outline theoretically and confirm experimentally up to $d = 15$ dimensions. Using the spatial modes of light as our encoding basis, we use this channel to transfer information expressed across many spatial bases, including OAM, Hermite-Gaussian and their superpositions. Our experiment is supported by a full theoretical treatment and offers a new approach to harnessing high-dimensional structured quantum states by nonlinear optical control and detection.

## Results
### Concept
A schematic of our concept is shown in Fig. 1 together with the experimental realisation in Fig. 2a, with full details provided in Supplementary Note 1. Two entangled photons, B and C, are produced from a nonlinear crystal ($NLC_1$) configured for collinear non-degenerate spontaneous parametric downconversion (SPDC). Photon C is sent to interact with the state to be transferred (coherent source A), as prepared using a spatial light modulator ($SLM_A$), while photon B is measured by spatial projection with a spatial light modulator ($SLM_B$) and SMF.

In our scheme, we overlap photons from the coherent source A with single photon C in a second nonlinear crystal ($NLC_2$), and detect the upconverted photon D, generated by means of sum frequency generation (SFG). The success of the process is conditioned on the measurement of the *single* photon D (due to the *single* photon C from the entangled pair) in coincidence with the *single* photon B from the entangled pair. We use a coherent state as input to enhance the probability for up-conversion, where all the photons carry the same modal information which we want to transport.

To understand the process better, it is instructive to use OAM modes as an example; a full basis-independent theoretical treatment is given in Supplementary Notes 2 through 4. We pump the SPDC crystal

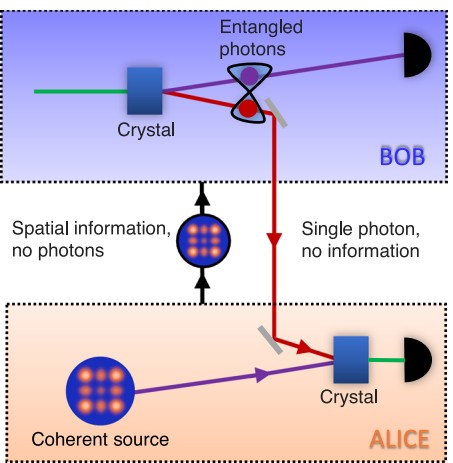

**Fig. 1 | High-dimensional quantum transport enabled by nonlinear detection.** In our concept, information is encoded on a coherent source and overlapped with a single photon from an entangled pair in a nonlinear crystal for up-conversion by sum frequency generation, the latter acting as a nonlinear spatial mode detector. The bright source is necessary to achieve the efficiency required for nonlinear detection. Information and photons flow in opposite directions: one of Bob's entangled photons is sent to Alice and has no information, while a measurement on the other in coincidence with the upconverted photon establishes the transport of information across the quantum link. Alice need not know this information for the process to work, while the nonlinearity allows the state to be arbitrary and unknown in dimension and basis.

with a Laguerre-Gaussian mode of azimuthal and radial indices $\ell_p = 0$ and $p_p = 0$, respectively. OAM is conserved in the SPDC process[15] so that $\ell_p = 0 = \ell_B + \ell_C$. The up-conversion process also conserves OAM[34], so if the detection is by a single mode fibre (SMF) that supports only spatial modes with $\ell_D = 0$, then $\ell_D = 0 = \ell_A + \ell_C$. One can immediately see that a coincidence is only detected when both A and B are conjugate to C, $\ell_A = \ell_B = -\ell_C$, and thus the prepared state (A) matches the transported state (B). One can show more generally (see Supplementary Note 2) that if the detection of photon D is configured to be into the same mode as the initial SPDC pump (we may call photon D the anti-pump), then the up-conversion process acts as the conjugate of the SPDC process, and the state of each photon in the coherent source A that is involved in the up-conversion is transported to that of photon B. To keep the language clear, we will refer to those photons in coherent source A that take part in the up-conversion as *photon-state A*, as in the SPDC process where only one pump photon is considered to take part in the down-conversion process, ignoring the vacuum term in both cases since they do not give rise to coincidences in our process. However, up-conversion aided quantum transport only takes place under pertinent experimental conditions, namely, perfect anti-correlations between the signal and idler photons from the SPDC process in the chosen basis, and an up-conversion crystal with length and phase-matching to ensure for anti-correlations between photon-state A and photon C (see Supplementary Note 3 for full details).

To find a bound on the modal capacity of the channel, one can treat the process as a communication channel with an associated channel operator. This, in turn, can be treated as an entangled state, courtesy of the Choi-Jamoilkowski state-channel duality[35], from which a Schmidt number ($K$) can be calculated. We interpret this as the effective number of modes the channel can transfer (its modal capacity), given by

$$K = \frac{\left[\int T^2(\mathbf{q}_A, \mathbf{q}_B) d^2\mathbf{q}_A d^2\mathbf{q}_B\right]^2}{\int \left[\int T(\mathbf{q}_A, \mathbf{q}_C) T(\mathbf{q}_C, \mathbf{q}_B) d^2\mathbf{q}_C\right]^2 d^2\mathbf{q}_A d^2\mathbf{q}_B}, \quad (1)$$

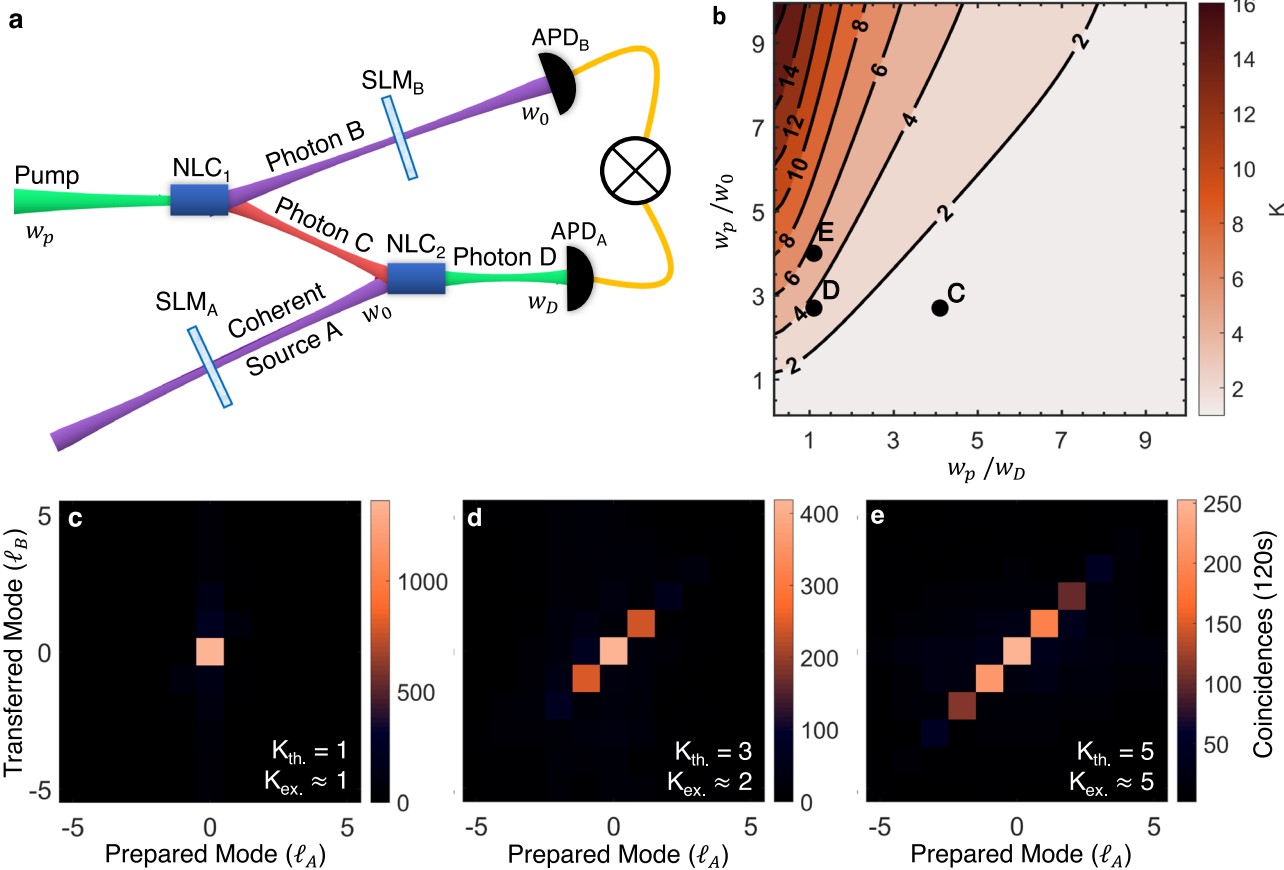

**Fig. 2 | Realising a quantum transport channel. a** A pump photon ($\lambda_p = 532$ nm) undergoes spontaneous parametric downconversion (SPDC) in a nonlinear crystal (NLC$_1$), producing a pair of entangled photons (signal B and idler C), at wavelengths of $\lambda_B = 1565$ nm and $\lambda_C = 808$ nm, respectively. Photon B is directed to a spatial mode detector comprising a spatial light modulator (SLM$_B$) and a single mode fibre coupled avalanche photo-diode detector (APD). The state to be transferred is prepared as a coherent source A using SLM$_A$ ($\lambda_A = 1565$ nm), and is overlapped in a second nonlinear crystal (NLC$_2$) with photon C, resulting in an upconverted photon D which is sent to a single mode fibre coupled APD. Photons B and D are measured in coincidence to find the joint probability of the prepared and measured states using the two SLMs. **b** The quantum transport channel's theoretical modal bandwidth ($K$) as a function of the pump ($w_p$) and detected photons' ($w_0$ and $w_D$) radii, with experimental confirmation shown in **c** through **e** corresponding to parameter positions C, D, and E in **b**. $K_{th}$ and $K_{ex}$ are the theoretical and experimental quantum transport channel capacities, respectively. The cross-talk plots are shown as orbital angular momentum (OAM) modes prepared and transferred. The raw data is reported with no noise suppression or background subtraction, and considering the same pump power conditions in all three configurations.

where

$$T(\mathbf{q}_A, \mathbf{q}_B) = \int \psi^*_{SFG}(\mathbf{q}_A, \mathbf{q}_C) \psi_{SPDC}(\mathbf{q}_C, \mathbf{q}_B) d^2\mathbf{q}_C,$$

with the SFG and SPDC wave functions expressed in the momentum (**q**) basis.

Full details are given in Supplementary Note 4. The controllable parameters are the beam radii of the pump ($w_p$), and the spatially filtered photons D ($w_D$) and B ($w_0$). Using Equation (1), we calculated the channel capacity for OAM modes, with the results shown in Fig. 2b, revealing that a large pump mode relative to the detected transferred modes is optimal for capacity. A large pump mode with respect to the crystal length also increases the channel capacity, consistent with the well-known thin-crystal approximation. However, this comes at the expense of coincidence events, the probability of detecting the desired OAM mode, which must be balanced with the noise threshold in the system. We show three experimental examples of this trade-off in Fig. 2c–e, where the parameters for each can be deduced from the corresponding labelled positions in Fig. 2b. Good agreement between theoretical ($K_{th}$) and experimentally measured ($K_{ex}$) capacities validates the theory. Using the theory, we adjust the experimental parameters to optimise the quantum transport channel, reaching a maximum of $K \approx 15$ for OAM modes, as shown in the inset of Fig. 3. This

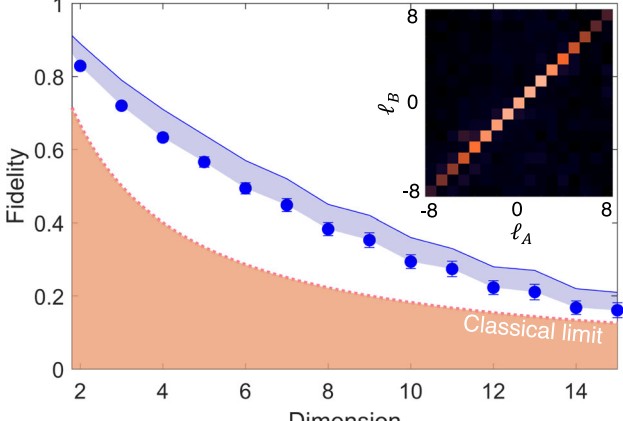

**Fig. 3 | Quality of the quantum transport process.** Experimental fidelities (points) for our channel dimensions up to the maximum achievable channel capacity of $K = 15 \pm 1$, all well above the classical limit (dashed line). The solid line forms a maximum fidelity for the measured transferred state. The inset shows the measured OAM modal spectrum of the optimised quantum transport channel with maximum coincidences of 320 per second for a 5 min integration time. The raw data is reported with no noise suppression or background subtraction.

limit is not fundamental and is set only by our experimental resources. We are able to establish a quantum transport setup where the channel supports at least 15 OAM modes. The balance of channel capacity with noise is shown in Fig. 3. Using a probe of purity and dimension[36] we use a traditional measure and estimate a channel fidelity which decreases with channel dimension, but is always well above the upper bound of the achievable fidelity for the classical case, i.e., having no entanglement between photons B and C, given by $F_{classical} \leq \frac{2}{d+1}$ for $d$ dimensions and shown as the classical limit (dashed line) in Fig. 3. Blue points show the quantum transport fidelity, measured from Eq. (12), using the channel fidelity $F_{Ch}$. Here, the channel fidelity measures the quality of the correlations that can be established between photon-state A and photon B over the two particle $d^2$ subspace while the quantum transport fidelity, $\mathcal{F}$ measures how well $SLM_B$ and $APD_B$ can measure states transmitted over the channel, requiring measurements over a single particle $d$ dimensional space. Since $F_{Ch} \leq \mathcal{F} \leq \frac{F_{Ch}d+1}{d+1}$, it follows that $\mathcal{F}$, shown as the solid line above the shaded region in Fig. 3, sets the upper-bound for the quantum transport fidelity[37] and is therefore the highest achievable fidelity for our system (See Methods for further details). Note that we use a measurement of a two particle system because we condition on coincidence events between *single* photons B and D.

## Quantum transport results

In Fig. 4 we show results for the quantum transport channel in two, three and four dimensions. We confirm quantum transport beyond just the computational basis by introducing a modal phase angle, $\theta$, on photon B relative to photon-state A ($\theta = 0$) for the two-dimensional state $|\Psi\rangle = |\ell\rangle + \exp(i\theta)|-\ell\rangle$ (we omit the normalization throughout the text for simplicity). We vary the phase angle while measuring the resulting coincidences for three example OAM subspaces, $\ell = \pm 1, \pm 2$ and $\pm 3$. The raw coincidences, without any noise subtraction, are plotted as a function of the phase angle in Fig. 4a, confirming the quantum transport across all bases. The resulting visibilities (V) allow us to determine the fidelities[38] from $F = \frac{1}{2}(1 + V)$, with raw values varying from 90% to 93%, and

background subtracted all above 98% (see Supplementary Notes 7 through 9). Example results for the qutrit state $|\Psi\rangle = |-1\rangle + |0\rangle + |1\rangle$ are shown in Fig. 4b as the real and imaginary parts of the density matrix, reconstructed by quantum state tomography, obtaining a transferred qutrit with an average channel fidelity of $0.82 \pm 0.016$ (see Supplementary Note 13 for all detailed measurements with the raw coincidences from the projections in all orthogonal and mutually unbiased basis). Further analysis of judiciously chosen transferred states themselves lead to even higher values (see Supplementary Notes 11 and 14).

Next, we proceed to illustrate the potential of the quantum transport channel by sending four-dimensional states of the form $|\Psi\rangle = |-3\rangle + \exp(i\theta_1)|-1\rangle + \exp(i\theta_2)|1\rangle + \exp(i\theta_3)|3\rangle$, with inter-modal phases of $\{\theta_1, \theta_2, \theta_3\} = \{-\pi/2, -\pi, -\pi/2\}, \{-\pi/2, 0, \pi/2\}, \{\pi/2, \pi, \pi/2\}$ and $\{\pi/2, 0, -\pi/2\}$. All possible outcomes from these mutually unbiased basis (MUBs) are shown in Fig. 4c. We encoded each superposition (one at a time) in $SLM_A$ and projected photon B in each of the four states. The strong diagonal with little cross-talk in the off-diagonal terms confirms quantum transport across all states. Figure 4d shows an exemplary detection of one such MUB state in the OAM basis: the transferred state (solid bars) with the prepared state (transparent bars), for a similarity of $S = 0.98 \pm 0.047$ (see description used in the Methods section). Note that the prepared states (transparent bars) in the figures throughout the letter are obtained by the averaged sum of all measured values involved, facilitating comparison with the raw coincidences. Furthermore, we have also transferred various unbalanced superpositions of OAM states (see Supplementary Note 12 and Suppl. Fig. 14 for full details), being able to assign different weightings. The encoded states are the following: $|\Psi\rangle = 2|-1\rangle + 3|0\rangle + |1\rangle$, $|\Psi\rangle = 2|-2\rangle + 3|0\rangle + |2\rangle$, $|\Psi\rangle = |-2\rangle + 2|0\rangle + |2\rangle$, and $|\Psi\rangle = 2|-3\rangle + |-1\rangle + |1\rangle + 2|4\rangle$.

The result in Fig. 4c also confirms that the channel is not basis dependent, since this superposition of OAM states is not itself an OAM eigenmode. To reinforce this message, we proceed to transfer $d = 3$ and $d = 9$ states in the Hermite-Gaussian ($HG_{n,m}$) basis with indices $n$ and $m$, with the results shown in Fig. 5.

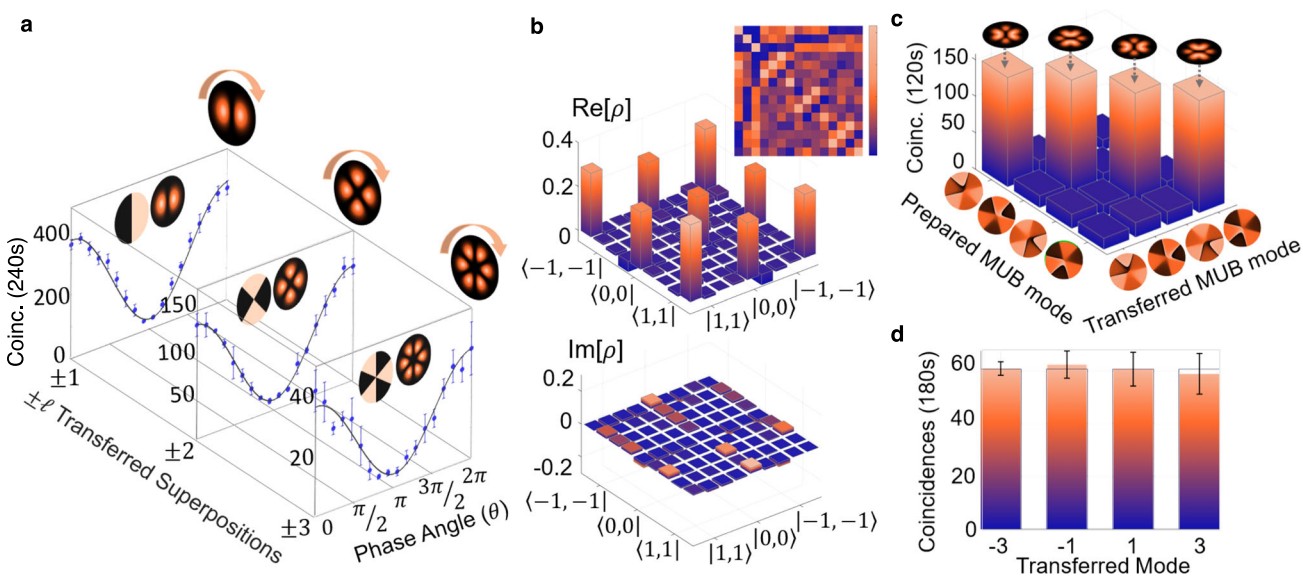

**Fig. 4 | Visibilities and quantum state tomography. a** Measured coincidences (points) and fitted curve (solid) as a function of the phase angle ($\theta$) of the corresponding detection analyser for the state $|\phi\rangle = |\ell\rangle + \exp(i\theta)|-\ell\rangle$, for three OAM subspaces of $\ell = \pm 1, \pm 2$, and $\pm 3$ (further details in the Supplementary Note 5). **b** The real (Re[$\rho$]) and imaginary (Im[$\rho$]) parts of the density matrix ($\rho$) for the qutrit state $|\Psi\rangle = |-1\rangle + |0\rangle + |1\rangle$ as reconstructed by quantum state tomography. The inset shows the raw coincidences with maximum coincidences of 220 detected per second from the tomographic projections (full details in the Supplementary Notes 6 and 13). **c** Measurements for the quantum transport of a 4-dimensional state, constructed from the states $\ell = \{\pm 1, \pm 3\}$. **d** Measurements showing the detection (solid bars) of all the prepared (transparent bars) OAM states comprising one of the MUB states. The raw data is reported with no noise suppression or background subtraction.

In both cases the measured state (solid bars) is in very good agreement with the prepared state (transparent bars). Note that the results only confirm that the diagonal terms of the density matrices of the input states were transported successfully and so cannot confirm the transportation of coherences (off-diagonal elements of the density matrices) before and after quantum transport. The good agreement between the diagonal elements of the initial and final states is evidence that the quantum transport works for these elements, corroborated by the full phase information already confirmed up to $d = 4$ and a channel capacity (that includes phases) up to $d = 15$. To quantify the final state's diagonal terms for $d = 9$ we make use of similarity as a measure (see Methods) because of the prohibitive time (due to low counts) to determine Fidelity from a quantum state tomography, but note that this measure does not account for modal phases in the prepared and measured state. A final summary of example transferred states is shown in Fig. 6, covering dimensions two through nine, and across many bases. The prepared (transparent bars) and transferred (solid bars) states are in good agreement, as determined from the similarity, confirming the quality of the channel. Note that the coincidence counts are given for the detected OAM states (solid bars). The weightings of the prepared ones (transparent bars) are intended to show the normalized probabilities for visualization purposes.

## Discussion

Structured quantum light has gained traction of late[39–41], promising a larger Hilbert space for information processing and communication. The use of nonlinear optics in the *creation* of high-dimensional quantum states is exhaustive (SPDC, photonic crystals, resonant meta-surfaces and so on), while the preservation of entanglement and coherence in nonlinear processes[42] has seen it used for efficient photon detection[43], particularly for measurement of telecom wavelength photons[44]. Full harnessing and controlling high-dimensional quantum states by nonlinear processes has however remained elusive. Notable exceptions include advances made in the time-frequency domain[45], another degree of freedom to harness high-dimensional states, such as the demonstration of quantum pulse gates[46] for efficient demultiplexing of temporal modes as well as for tomographic measurements[47,48], the inverse process of multiplexing by difference frequency generation[49], quantum interference of spectrally distinguishable sources[50], high-dimensional information encoding[51] and simultaneous temporal shaping and detection of quantum wavefunctions[52]. To the best of our knowledge, our work is the first in the spatial domain, offering an exciting resource for controlling and processing spatial quantum information by nonlinear processes. Combining advances in high-dimensional spectral-temporal state control[53] and on-chip nonlinear solutions[54] with the spatial degree of freedom could herald new prospects in quantum information processing beyond qubits.

In conclusion, we have demonstrated an elegant way to perform a projection of an unknown state using a nonlinear detector, facilitating quantum information in high dimensions, and across many spatial bases, to be transferred with just one entangled pair as the quantum resource. Our results validate the non-classical nature of the channel without any noise suppression or background subtraction. While our quantum transport scheme cannot teleport entanglement due to the need of encoding the state to be transferred in many copies, it nevertheless securely transfers the state of the laser photons to the distant and previously entangled photon, and it does this without using knowledge of the state of the laser photons (see Supplementary Notes 10 and 15 discussing the challenges to move from transport to

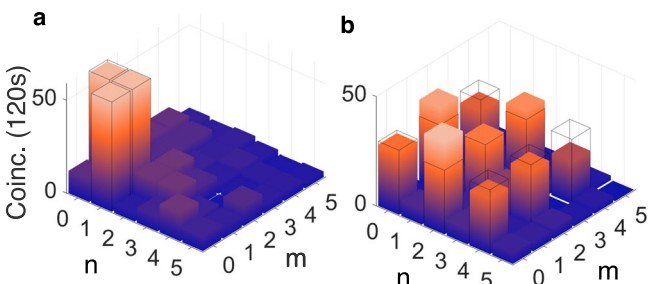

**Fig. 5 | Quantum transport in the Hermite-Gaussian basis.** Coincidence measurements for quantum transport of **a** a 3-dimensional and **b** a 9-dimensional HG$_{n,m}$ state, constructed from the states $(n, m) = \{(0, 1), (1, 0), (1, 1)\}$ and $(n, m) = \{(0, 0), (2, 0), (0, 2), (2, 2), (2, 4), (4, 2), (4, 4)\}$, respectively. The weights of the diagonal elements of the density operator of the transported state (solid bars) are in good agreement with the weights of the prepared state (transparent bars). The raw data is reported with no noise suppression or background subtraction.

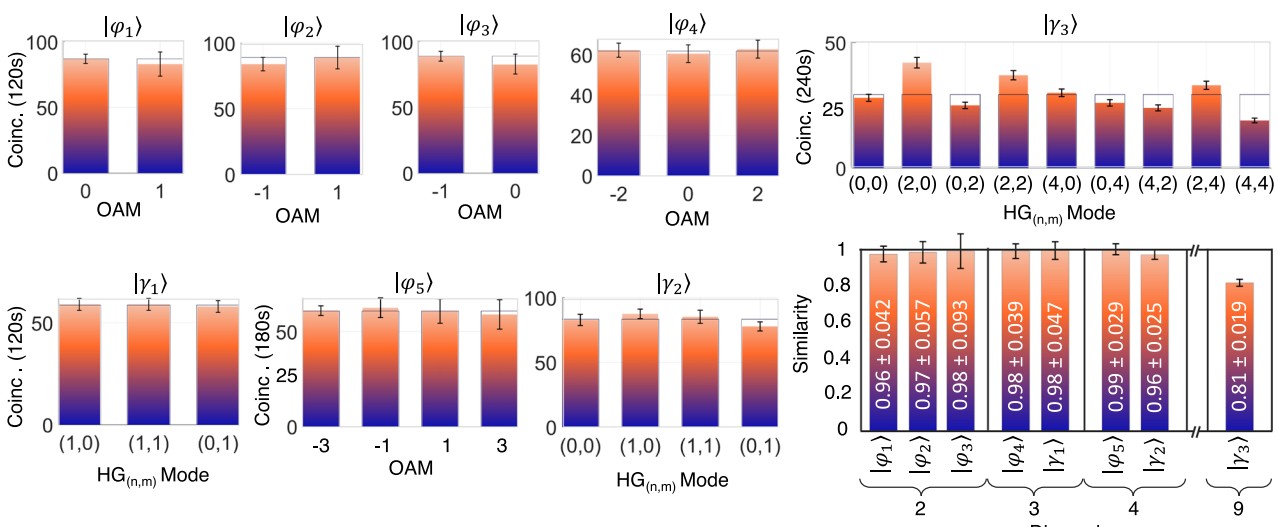

**Fig. 6 | Summary of transferred states.** Similarities for transport of a 2,3,4 and 9-dimensional superposition states in the OAM (represented as $|\varphi\rangle$) and HG (represented as $|\gamma\rangle$) bases shown and labelled to the left. Transferred states are $|\varphi_1\rangle = |0\rangle + |-1\rangle$, $|\varphi_2\rangle = |-1\rangle + |1\rangle$, $|\varphi_3\rangle = |0\rangle - |1\rangle$, $|\varphi_4\rangle = |-2\rangle + |0\rangle + |2\rangle$, $|\gamma_1\rangle = |HG_{1,0}\rangle + |HG_{1,1}\rangle + |HG_{0,1}\rangle$, $|\varphi_5\rangle = |-3\rangle - i|-1\rangle + |1\rangle + i|3\rangle$, $|\gamma_2\rangle = |HG_{0,0}\rangle + |HG_{1,0}\rangle +$ $|HG_{1,1}\rangle + |HG_{0,1}\rangle$ and $|\gamma_3\rangle = |HG_{0,0}\rangle + |HG_{2,0}\rangle + |HG_{0,2}\rangle + |HG_{2,2}\rangle + |HG_{4,0}\rangle +$ $|HG_{0,4}\rangle + |HG_{4,2}\rangle + |HG_{2,4}\rangle + |HG_{4,4}\rangle$. The similarity of diagonal elements of the density matrix together with prior phase information confirms coherent transport up to $d = 4$ but not for $d = 9$, where only the diagonal elements are assessed. Raw data are reported without noise suppression or background subtraction.

teleport). Importantly, our comprehensive theoretical treatment outlines the tuneable parameters that determine the modal capacity of the quantum transport channel, such as modal sizes at the SPDC crystal and detectors, requiring only minor experimental adjustments (for example, the focal length of the lenses). The modal capacity of our channel was limited only by experimental resources, while future research could target an increase of the number of transferred modes by optimising the choice of the relevant parameters and improved nonlinear processes. Our work highlights the exciting prospect this approach holds for the quantum transport of unknown high-dimensional spatial states, and could in the future be extended to mixed degrees of freedom, for instance, hybrid entangled (polarization and space) and hyper entangled (space and time) states, for multi-degree-of-freedom and high-dimensional quantum control.

## Methods
### Fidelity
To quantify the quality of the quantum transport process, we use *fidelity*. It is defined for pure states as the squared magnitude of the overlap between the initial state that was to be transferred $|\psi_A\rangle$ and the final transferred state that was received by SLM$_B$ and APD$_B|\psi_B\rangle$:

$$F = |\langle \psi_A | \psi_B \rangle|^2. \tag{2}$$

In the ideal case, where the transferred state is $|\psi_B\rangle = \int \alpha(\mathbf{q}) \hat{a}_B^\dagger(\mathbf{q}) | \text{vac} \rangle d\mathbf{q}$ (with detailed description in Supplementary Note 2), the fidelity is $F = 1$. However, in a practical experiment, the conditions for the ideal case cannot be met exactly. Therefore, the fidelity is given more generally by

$$
\begin{aligned}
F &= \int \alpha^*(\mathbf{q}_B) \beta(\mathbf{q}_B) \, d^2 q_B \\
&= \int \alpha^*(\mathbf{q}_B) U^*(\mathbf{q}_D) g(\mathbf{q}_A, \mathbf{q}_D - \mathbf{q}_A, \mathbf{q}_D) \\
&\quad \times f(\mathbf{q}_B, \mathbf{q}_D - \mathbf{q}_A) \alpha(\mathbf{q}_A) \, d^2 q_B \, d^2 q_A \, d^2 q_D.
\end{aligned}
\tag{3}
$$

Here, $f(\mathbf{q}_B, \mathbf{q}_C)$ is the two photon wave-function of the SPDC state, while $g(\mathbf{q}_A, \mathbf{q}_C, \mathbf{q}_D)$ and $U(\mathbf{q}_D)$ are the SFG kernel and projection mode for photon $D$ (the up-converted photon), respectively. It is possible to envisage a classical implementation of the state-transfer process. One would make a complete measurement of the initial state, send the information and then prepare photon B with the same state. To ensure that the quantum transport process can outperform this classical state-transfer process, the fidelity of the process must be better than the maximum fidelity that the classical quantum transport process can obtain.

In order to determine the classical bound on the fidelity by which we measure the transferred state $|\psi\rangle$, we define the probability of measuring a value $a$ by

$$P_\psi(a) = \langle \psi | \hat{E}_a | \psi \rangle, \tag{4}$$

where $\hat{E}_a$ is an element of the positive operator valued measure (POVM) for the measurement of the initial state. These elements obey the condition.

$$\sum_a \hat{E}_a = \mathbb{I}, \tag{5}$$

where $\mathbb{I}$ is the identity operator. The estimated state associated with such a measurement result is represented by $|\psi_a\rangle$.

For the classical bound, we consider the average fidelity that would be obtained for all possible initial states. This average fidelity is

given by

$$
\begin{aligned}
\mathcal{F} &= \int \sum_a P_\psi(a) |\langle \psi | \psi_a \rangle|^2 \, d\psi, \\
&= \int \sum_a \langle \psi | \hat{E}_a | \psi \rangle |\langle \psi | \psi_a \rangle|^2 \, d\psi,
\end{aligned}
\tag{6}
$$

where $d\psi$ represents an integration measure on the Hilbert space of all possible input state. We assume that this space is finite-dimensional but larger than just two-dimensional. Since all the states in this Hilbert space are normalized, the space is represented by a hypersphere. A convenient way to represent such an integral is with the aid of the Haar measure. For this purpose, we represent an arbitrary state in the Hilbert space as a unitary transformation from some fixed state in the Hilbert space $|\psi\rangle \to \hat{U}|\psi_0\rangle$, so that $d\psi \to dU$. The average fidelity then becomes the following

$$
\begin{aligned}
\mathcal{F} &= \int \sum_a \langle \psi_a | \hat{U} | \psi_0 \rangle \langle \psi_0 | \hat{U}^\dagger \hat{E}_a \hat{U} | \psi_0 \rangle \langle \psi_0 | \hat{U}^\dagger | \psi_a \rangle \, dU \\
&= \int \sum_a \text{tr} \{ \hat{\rho}_a \hat{U} \hat{\rho}_0 \hat{U}^\dagger \hat{E}_a \hat{U} \hat{\rho}_0 \hat{U}^\dagger \} \, dU.
\end{aligned}
\tag{7}
$$

The general expression for the integral of the tensor product of four such unitary transformations, represented as matrices, is given by

$$
\begin{aligned}
&\int U_{ij}(U^\dagger)_{kl} U_{mn}(U^\dagger)_{pq} \, dU \\
&= \frac{1}{d^2 - 1} \left( \delta_{il}\delta_{jk}\delta_{mq}\delta_{np} + \delta_{iq}\delta_{jp}\delta_{ml}\delta_{nk} \right) \\
&\quad - \frac{1}{(d^2 - 1)d} \left( \delta_{il}\delta_{jp}\delta_{mq}\delta_{nk} + \delta_{iq}\delta_{jk}\delta_{ml}\delta_{np} \right).
\end{aligned}
\tag{8}
$$

Using this result in Eq. (7), we obtain

$$\mathcal{F} = \frac{1}{(d+1)d} \left( d + \sum_a \langle \psi_a | \hat{E}_a | \psi_a \rangle \right), \tag{9}$$

where $d$ is the dimension of the Hilbert space and where we imposed $\text{tr} \{\hat{\rho}_0\} = \text{tr} \{\hat{\rho}_0^2\} = \text{tr} \{\hat{\rho}_a\} = 1$. We see that $\mathcal{F}$ is maximal if $E_a$ represents rank 1 projectors and $E_a |\psi_a\rangle = |\psi_a\rangle$, that is, $E_a = |\psi_a\rangle\langle\psi_a|$. Then $\sum_a \langle \psi_a | \hat{E}_a | \psi_a \rangle = d$. It follows that the upper bound of the fidelity achievable for the classical state-transfer process is given by[37]

$$\mathcal{F} \leq \frac{2}{d+1}. \tag{10}$$

The fidelity obtained in quantum transport needs to be better than this bound to outperform the classical scheme.

Furthermore, we can consider the particular quantum transport of a subspace smaller than the supported by the quantum transport channel capacity (see more details in Supplementary Note 4). The quantum transport fidelity of the channel for each subspace $d'$ within the $d$ dimensional state, $\rho$, can be computed by truncating the density matrix and overlapping it with a channel state that has perfect correlations. The theoretical fidelity is given by the expression[37]

$$F_{Ch} = \frac{d'(p' - 1) + d'^2}{d'^2}, \tag{11}$$

where $p'$ and $d'$ are the purity and dimensionality of truncated states. While this assumes that the channel has a random noisy component given by $\mathbb{I}_{d'^2}/d'^2$, the photon C only has a noise component given by $\mathbb{I}_{d'}/d'$ therefore the quantum transport fidelity for each photon is

given by,

$$\mathcal{F} = \frac{F_{Ch}d' + 1}{d' + 1}. \tag{12}$$

Here the separability criterion admits the classical bounds $\frac{1}{d'^2} \leq F_{Ch} \leq \frac{1}{d'}$ and $\frac{1}{d'} \leq \mathcal{F} \leq \frac{2}{d'+1}$ for the full channel and a single state received, respectively.

## Similarity

We use a normalised distance measure to quantify the quality of the state being transferred, denoted the *Similarity* (S),

$$S = 1 - \frac{\sum_j |(|C_j^{Ex.}|^2 - |C_j^{Th.}|^2)|}{\sum_j |C_j^{Ex.}|^2 + \sum_j |C_j^{Th.}|^2}. \tag{13}$$

Here we take the normalised intensity coefficients, $|C_j^{Th.}|^2$, encoded onto $SLM_A$ for the *j*th basis mode comprising the state being transferred (i.e., $|\Phi\rangle = \sum_j C_j |j\rangle$) and compare it with the corresponding *j*th coefficient $|C_j^{Ex.}|^2$ detected after traversing the quantum transport channel (made with *j*th-mode projections on $SLM_B$) as described in the Supplementary Note 4. A small difference in values between encoded and detected state would result in a small 'distance' between the prepared and received value. As such, the second term in Eq. (13) diminishes with increasing likeness of the states, causing the Similarity measure to tend to 1 for unperturbed quantum transport of the state.

## Dimensionality measurements

We employ a fast and quantitative dimensionality measure to determine the capacity of our quantum channel. The reader is referred to ref. 36 for full details, but here we provide a concise summary for convenience. The approach coherently probes the channel with multiple superposition states $|M,\theta\rangle_n$.

We construct the projection holograms from the states

$$U_n(\phi,\theta) = \mathcal{M} \sum_{k=0}^{n-1} \exp(i\Phi_M(\phi; \beta_k \oplus \theta)), \tag{14}$$

which are superpositions of fractional OAM modes,

$$\exp(i\Phi_M(\phi;\theta)) = \begin{cases} e^{iM(2\pi+\phi-\theta)} & 0.5em 0 \leq \phi < \theta \\ e^{iM(\phi-\theta)} & 0.5em \theta \leq \phi < 2\pi \end{cases}, \tag{15}$$

s rotated by an angle $\beta_k \oplus \theta = \mod\{\beta_k + \theta, 2\pi\}$ for $\beta_k = \frac{2\pi}{n}k$. Here, $\phi$ is the azimuthal coordinate.

While $\theta$ determines the relative phase for the projections, physically it corresponds to the relative rotation of the holograms. After transmitting the photon imprinted with the state $|M,\theta\rangle_n$, through the quantum transport channel, $\hat{T} = \sum_\ell \lambda_\ell |\ell\rangle_A \langle\ell|_B$, the photon is projected onto the state $|M,0\rangle_n$. The detection probability is then given by

$$P_n(\theta) = |\langle 0,M|\hat{T}|M,\theta\rangle|^2, \tag{16}$$

having a peak value at $P(\theta = 0)$ and a minimum at $P(\theta = \pi/n)$. In the experiment, there are noise contributions which can be attributed to noise from the environment, dark counts and from the down-conversion and up-conversion processes. Since the channel is isomorphic to an entangled state, i.e

$$\hat{T} = \sum_\ell \lambda_\ell |\ell\rangle_A \langle\ell|_B \rightarrow \rho_{\hat{T}} := \sum_\ell \lambda_\ell |\ell\rangle_A |\ell\rangle_B, \tag{17}$$

we represent the system by an isotropic state,

$$\rho = p\rho_{\hat{T}} + (1-p)\mathbb{I}_d^2/d^2, \tag{18}$$

where $p$ is the probability of transferring a state through the channel or equivalently the purity and $\mathbb{I}_d^2$ is a $d^2$ dimensional identity matrix. In this case, the detection probability is given by

$$P_n(\theta) = |\langle 0,M|\hat{T}|M,\theta\rangle|^2 + (1-p)/d^2 I_n(\theta), \tag{19}$$

where $I_n(\theta) = |\langle 0,M|\mathbb{I}_{d^2}|M,\theta\rangle|^2$.

We compute the visibilities

$$V_n = \frac{|P_n(0) - P_n(\pi/n)|}{|P_n(0) + P_n(\pi/n)|}. \tag{20}$$

Using the fact that the visibility, $V_n := V_n(p,d)$, obtained for each analyser indexed by, $n = 1, 3, \ldots, 2N-1$, scales monotonically with $d$ and $p$[36], we determine the optimal $(p,d)$ pair that best fit the function $V_n(p,K)$ to all $N$ measured visibilities by employing the method of least squares (LSF). The fidelity for the channel, $F_{Ch}$, can therefore be computed by overlapping the truncated subspaces of dimensions $d'$ in the $d$ dimensional state from Eq. (18), with a channel state having perfect correlations. From this we compute the quantum transport fidelity, $\mathcal{F}$ from Eq. (12).

## Data availability

The data that supports the plots within this paper and other findings of this study are available from the corresponding authors upon request.

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

## Acknowledgements

B.S. would like to acknowledge the Department of Science and Innovation and Council for Industrial and Scientific Research (South Africa) for funding. A.V. acknowledges the MCIN with funding from European Union (QSNP, 101114043), Next Generation EU (PRTR-C17.I1), and the Japan Society for the Promotion of Science for funding (JSPS-KAKENHI - G21K14549). F.S. acknowledges financial support by the Fraunhofer Internal Programs under Grant No. Attract 066-604178. M.A.C., F.S.R., and A.F. thanks the National Research Foundation for funding (NRF Grant No. 121908, 118532, TTK2204011621). A.V. and J.P.T. acknowledge financial support from the "Severo Ochoa" program for Centres of Excellence CEX2019-000910-S [MICINN/ AEI/10.13039/501100011033], Fundació Cellex, Fundació Mir-Puig, and Generalitat de Catalunya through CERCA, from project 20FUN02 "POLight" funded by the EMPIR programme, and from project QUISPAMOL (PID2020-112670GB-I00).

## Author contributions

The experiment was performed by B.S., A.V. and I.N., with technical support by M.A.C. and the theory developed by F.S., T.K., J.P.T. and F.S.R. Data analysis was performed by B.S., A.V., I.N. and A.F. and the experiment was conceived by A.V., F.S., T.K., J.P.T., F.S.R. and A.F. All authors contributed to the writing of the manuscript. A.F. supervised the project.

## Competing interests

The authors declare no competing interests.

## Additional information

**Peer review information** : *Nature Communications* thanks the anonymous reviewer(s) for their contribution to the peer review of this work. A peer review file is available.

