## [Peer Review File · Nature Communications]

Quantum transport of high-dimensional spatial information with a nonlinear detectorREVIEWER COMMENTS

Reviewer #1 (Remarks to the Author):

The authors report an experiment, in which they transport a high-dimensional quantum state, encoded in the spatial mode of a bright coherent laser field, from the laser to the spatial mode of a single photon using a nonlinear process. To achieve this quantum transport (as dubbed by the authors), one photon out of the laser is upconverted by mixing it with another photon in an SFG process and subsequently detected in post-selection. The trick is that the photon that is mixed with the laser was initially entangled with the photon to which the quantum state is transferred. In other words, the authors developed a nonlinear detection scheme, which can be controlled through the shaping of the spatial mode of the laser, thereby projecting one photon out of a high-dimensionally entangled pair onto a specific mode. This projection is then – through the virtue of entanglement – also found on the entangled partner. If the bright laser would be replaced by a single photon (which is impossible with current technologies) the presented scheme would correspond to a high-dimensional quantum teleportation scheme.

I really enjoyed reading the manuscript, as it is well written with enough details in the main text to follow the obtained results. In addition, the manuscript is accompanied by a supplementary material which includes all experimental and theoretical details to an extend that is rarely found. Although it is questionable if a nonlinear process will ever be developed/found that is efficient enough to make the scheme work as a teleportation scheme (only time will tell), I have to say that even in its current form is a nice step forwards and will have impact on the current high-dimensional quantum optics research. The results are nicely presented and of high quality, despite being a challenging experiment in its first implementation. In addition, the theory is described in detailed and convincing. As such, I can suggest publication as it is and have only minor comments that can be found below.

Comments:

- Page 1: the ref for ion teleportation is out of place. The following sentence starting with “A large component...” sounds as if the earlier referenced work already realized high-dimensional teleportation (which is not the case) or what do I misunderstand?
- Page 2, left column, last paragraph: “while photon B is projected onto a spatial light

modulator...” There is something wrong, the SLM in connection with the SMF projects the photon onto a state!

- Page 2, right column, last sentence second paragraph starting with “One can consider...”: What do the authors mean by the final part “coherent superposition”. This doesn’t make sense to me?

- Page 4, right column, first paragraph: The authors mention that the inset shows an example state. Which inset do the authors mean?

- Page 7, methods section, fidelity: the word “prepare” is appearing two times in a row

- Page 8, methods section, dimensionality measurements: I was a bit lost in the first part of this section so maybe the authors want to go through it and try to clarify it, e.g. I don’t see what states equation (14) describe? I only see a sum of transverse phases? Why do the authors use fractional OAM states here? I assume that this is just another way of writing OAM superpositions states, but it is directly clear to me?

- Page 24: The authors give mean values and experimentally obtained standard deviations, however, they miss to say how the standard deviations are obtained? I assume that they perform 1second long measurements as the whole measurement runs are given, but this should be stated somewhere.

- Page 25, section XIV and Fig.25: the authors study the difference between a flat spiral spectrum and the one they obtained in the experiment. The former would lead to a better result, which is nice to see. However, in section VI, the authors say that they use procrustean filtering such that the detected spectrum is indeed flat. What did the authors use in the experiment and is section XIV then still valid?

Reviewer #2 (Remarks to the Author):

Sephton and co-authors demonstrate a non-linear technique to project arbitrary high-dimensional states using information encoded in a bright coherent state and an entangled pair of photons as a quantum resource. This is presented as the exchange of high-dimensional information between distant locations, without direct transmission of the physical system containing the information, where the information to transmit is unknown to the sender. The capabilities of the technique are studied with information encoded in different spatial bases and dimensionalities. The manuscript presents an exhaustive study of

tuneable parameters that determine the modal capacity of the channel to demonstrate the potential of the scheme for the transport of spatial information in dimensions up to 15. However, there are significant problems with the conceptualisation, methodology and conclusions of this work.

The authors motivate their 'quantum transport' scheme as a technical solution to the fundamental and technological limitations of quantum teleportation of high-dimensional states, where linear schemes require ancillary photons and non-linear schemes are hindered by extremely low efficiencies. To justify that their transport scheme is based on the same physical principles as up-conversion-aided quantum teleportation (Ref. [42, 58]), the authors describe the state to be transferred as that of a single photon A. This photon is one of the many identical photons of the strong coherent field where information is encoded. On page 6, it is mentioned that although the state to be transferred is not destroyed due to the existence of multiple copies, Alice cannot use them. To this purpose, they introduce Charlie as a party that prepares the coherent state, while Alice (the sender) merely "directs photon A and photon B to the non-linear crystal". This suggests that Charlie can select a single photon of the strong laser field and send it to Alice, who somehow can only access this photon and not the rest of the coherent field. Meanwhile, the large number of photons in the coherent state act as an efficiency enhancer in the measurement that Alice performs. This protocol is contradictory and does not describe the experiment, which requires a different theoretical treatment to the one presented. For example, to establish an appropriate definition of selecting or removing one photon from a coherent field should consider that a coherent state implies the existence of a probability where the photon occupation number is zero. Even more, substituting the single photon field with a coherent state raises important concerns about the security of the protocol and its superiority over classical methods. The introduction of Charlie in the protocol seems like an attempt to conceal the fact that in any practical scenario, an eavesdropper could easily tomograph the coherent state and extract the information to be transmitted. This issue must be addressed in the text, especially considering claims such as "our experiment is based on the same physics, utilizing entanglement and classical communication to securely transmit a quantum state from one system to another" or "it securely transmits the state of the laser photons to the distant and previously entangled photon".

I also find problematic the comparison of the fidelity of the transfer against the classical

case, i.e., having no entanglement between photon B and C. The authors consider a classical bound that assumes only one copy of the state (Phys. Lett. A 253, 249-251 (1999), Ref. [52], Phys. Rev. Lett. 72, 797 (1994), and Phys. Rev. Lett. 74, 1259 (1995)), but the protocol uses a coherent state to encode the information (i.e., there are many copies of the state in question). As such, there doesn't seem to be an advantage over the case of Alice using all the available multiple copies to accurately and efficiently tomograph the state and send the result to Bob through a classical channel. The fact that a quantum resource such as entanglement can be used, does not necessarily mean that there is an advantage over the classical counterpart.

With all of this, I cannot recommend publishing the paper in its current state because it presents the work as a "versatile and scalable" alternative to exchanging high-dimensional information through quantum teleportation. This comparison is inaccurate because the two schemes have conceptual differences, they don't provide the same communication advantages and have very different relevance in the field of quantum technologies. The implication of introducing quantum transport as a strategy that "converges to high-dimensional quantum teleportation in the limit of a single photon input" misleads the reader on its relevance and application, undermining the significance of any previous or future demonstrations of high-dimensional quantum state teleportation. The authors should reformulate the theoretical framework in the "Experimental concept" section and Supplementary Notes II and X, addressing the problems I've pointed out and being clear about the use of a coherent field in the experimental setup in Fig. 1.a. They also need to indicate the existence and accessibility of the many copies of the state, and how they invalidate the transport properties mentioned in the last paragraph on page 5. Furthermore, they must change the narrative in the section "From transport to teleport" and remove direct comparisons to the quantum teleportation results of Ref. [45-46] on page 4 (they incorrectly claim to have replicated the state-of-the-art results in both two and three dimensions), and in Supplementary Note XI (benchmarking the transport protocol by comparing it to high-dimensional teleportation demonstrations does not make sense). After executing these changes, the study could still be of relevance to the field. The experiment demonstrates how non-linear parametric processes allow for arbitrary high-dimensional state projections in the spatial degree of freedom, where a strong coherent field enhances the probability of the process to efficiently characterize the operation and

probe its limitations. The technique could be put in context when comparing it with measurement strategies in the time-frequency domain that use SFG for the selection of arbitrary temporal modes (See the Quantum Pulse Gate [Opt. Express 19, 13770 (2011), Phys. Rev. A 90, 030302 (2014)], or the coherent projections on time-bin encoded photons [PRL 111, 153602 (2013)]). Furthermore, studying the spatial mode capabilities of the up-conversion channel could prove useful when advancements in up-conversion efficiencies permit real demonstrations of high-dimensional quantum teleportation through non-linear schemes.

However, some important technical aspects require attention:

- 1) For the two-dimensional state, coherent transport is shown with the introduction of a modal phase angle between photons A and B and the calculation of fringe visibilities. On the other side, three-dimensional transport is confirmed with a fidelity estimation through state tomography. Since a relationship between the visibilities and the state fidelity is not provided, a comparison of the transfer in these two different dimensions can't be made. The authors should then calculate the fidelity for the two-dimensional case.
- 2) Tomographic measurements support three-dimensional transfer. However, results in dimensions greater than three rely on a metric called similarity. This figure of merit is not standard in quantum state characterisation, and it doesn't demonstrate a coherent transfer of the state because it lacks phase information, a crucial aspect of quantum communication schemes. The authors need to calculate the state fidelity in the four and nine-dimensional cases, either with complete tomographic measurements or with the use of a witness. Otherwise, any claim of quantum transport beyond dimension three should be removed.
- 3) The results in Fig. 3.c) and Fig. 5 only show diagonal terms of the crosstalk matrix, which convey the invalid notion that they are sufficient to confirm the "agreement" between encoded spatial information and the projected state. These results could also describe the transfer of a maximally mixed state. The authors should include full crosstalk measurements between prepared and measured states to support their claims of quantum transport.
- 4) In Figs. 3-5, transparent bars are said to describe the prepared states. However, what do coincidence counts of the prepared state even mean?

Additionally, please have a careful read of the manuscript and correct the typos. For example:

- In the abstract: scaleable.
- In the caption for Fig. 1: “respectively. and The cross-talk plots”
- On page 6, last paragraph: securily
- Supplementary Note IX, Quality of the entanglement channel: “projected onto in the the sender arm”

RESPONSE TO THE REVIEWER COMMENTS

Reviewer #1 (Remarks to the Author):

The authors report an experiment, in which they transport a high-dimensional quantum state, encoded in the spatial mode of a bright coherent laser field, from the laser to the spatial mode of a single photon using a nonlinear process. To achieve this quantum transport (as dubbed by the authors), one photon out of the laser is upconverted by mixing it with another photon in an SFG process and subsequently detected in post-selection. The trick is that the photon that is mixed with the laser was initially entangled with the photon to which the quantum state is transferred. In other words, the authors developed a nonlinear detection scheme, which can be controlled through the shaping of the spatial mode of the laser, thereby projecting one photon out of a high-dimensionally entangled pair onto a specific mode. This projection is then – through the virtue of entanglement – also found on the entangled partner. If the bright laser would be replaced by a single photon (which is impossible with current technologies) the presented scheme would correspond to a high-dimensional quantum teleportation scheme.

I really enjoyed reading the manuscript, as it is well written with enough details in the main text to follow the obtained results. In addition, the manuscript is accompanied by a supplementary material which includes all experimental and theoretical details to an extent that is rarely found. Although it is questionable if a nonlinear process will ever be developed/found that is efficient enough to make the scheme work as a teleportation scheme (only time will tell), I have to say that even in its current form is a nice step forwards and will have impact on the current high-dimensional quantum optics research. The results are nicely presented and of high quality, despite being a challenging experiment in its first implementation. In addition, the theory is described in detailed and convincing. As such, I can suggest publication as it is and have only minor comments that can be found below.

We would like to thank the reviewer for the very good summary of what we have done and for the kind appraisal of our work. It was indeed a very challenging experiment and we are sure it will stimulate further research into the topic of quantum nonlinear optics with structured photons.

Comments:

- Page 1: the ref for ion teleportation is out of place. The following sentence starting with “A large component...” sounds as if the earlier referenced work already realized high-dimensional teleportation (which is not the case) or what do I misunderstand?

Thank you, this is indeed not true. But now we have removed the teleportation emphasis so the structure of the introduction is altered.

- Page 2, left column, last paragraph: “while photon B is projected onto a spatial light modulator...” There is something wrong, the SLM in connection with the SMF projects the photon onto a state!

Yes, indeed you are correct! This was a silly typo (not “onto” but “with”) and has been rephrased correctly: projected onto a state with an SLM and SMF.

- Page 2, right column, last sentence second paragraph starting with “One can consider...”: What do the authors mean by the final part “coherent superposition”. This doesn’t make sense to me?

Thank you for the query. Yes, something was lost here and it now reads that the coherent state input can be considered as a carrier of the spatial information to be transported.

- Page 4, right column, first paragraph: The authors mention that the inset shows an example state. Which inset do the authors mean?

Well spotted. We have redrawn Fig. 3(c) and fixed the main text accordingly. We have separated it now into two plots, (c) and (d), for a better understanding.

- Page 7, methods section, fidelity: the word “prepare” is appearing two times in a row

Thank you, corrected.

- Page 8, methods section, dimensionality measurements: I was a bit lost in the first part of this section so maybe the authors want to go through it and try to clarify it, e.g. I don’t see what states equation (14) describe? I only see a sum of transverse phases? Why do the authors use fractional OAM states here? I assume that this is just another way of writing OAM superpositions states, but it is not directly clear to me?

Thank you for this. What we have done is provide a reference to the original work which was published in this very journal (we did reference it in the main text as old ref [51]). We put the approach in the Methods for the benefit of the reader. We have added a brief introduction and relocated the reference now to help the reader to follow the method.

- Page 24: The authors give mean values and experimentally obtained standard deviations, however, they miss to say how the standard deviations are obtained? I assume that they perform 1second long measurements as the whole measurement runs are given, but this should be stated somewhere.

Thank you, we have now added this information to the text in the Supplementary Note 13. We repeated the measurements between three and five times (limited by time constraints due to long acquisition times) and from this computed the average and standard deviation. A propagation of error analysis was then used to obtain the corresponding uncertainties in all the preceding values and measures that were computed.

- Page 25, section XIV and Fig.25: the authors study the difference between a flat spiral spectrum and the one they obtained in the experiment. The former would lead to a better result, which is nice to see. However, in section VI, the authors say that they use procrustean filtering such that the detected spectrum is indeed flat. What did the authors use in the experiment and is section XIV then still valid?

Thank you for bringing this to our attention. We have now added a sentence in the Supplementary Note 13 specifying where this spiral flattening technique was used. We applied the fixed and characterised filtering value to compensate for changes in the efficiencies of individual OAM modes, only in some cases where we prepared several modal superpositions, such as the top detection graph of old Fig. 3(c) (Fig. 4 (d) now), larger OAM superposition states in old Fig. 5 (Fig. 6 now), and Suppl. Fig. 19. Conversely, one may use a basis, such as the MUB cross-talk detection matrix in Fig. 4(c), where such compensation is not necessary. On the other hand, for OAM = 0 and +-1 (qutrit), the coupling sizes ensured flat detection for these states. As a result, such filtering was not necessary in these cases or the tomography in Fig. 4(b). The plot in Suppl. Fig. 25 section XIV is still valid because this shows how the channel performs with a standard SPDC spectrum and a flat spectrum, but what you send across the channel is up to you.

Reviewer #2 (Remarks to the Author):

Sephton and co-authors demonstrate a non-linear technique to project arbitrary high-dimensional states using information encoded in a bright coherent state and an entangled pair of photons as a quantum resource. This is presented as the exchange of high-dimensional information between distant locations, without direct transmission of the physical system containing the information, where the information to transmit is unknown to the sender. The capabilities of the technique are studied with information encoded in different spatial bases and dimensionalities. The manuscript presents an exhaustive study of tuneable parameters that determine the modal capacity of the channel to demonstrate the potential of the scheme for the transport of spatial information in dimensions up to 15.

Thank you for the concise and accurate summary. In what follows we hope to convince you that we have made a concerted effort to address all your queries to place the work fairly in the context of others, and to highlight the advance that this work represents.

However, there are significant problems with the conceptualisation, methodology and conclusions of this work. The authors motivate their 'quantum transport' scheme as a technical solution to the fundamental and technological limitations of quantum teleportation of high-dimensional states, where linear schemes require ancillary photons and non-linear schemes are hindered by extremely low efficiencies. To justify that their transport scheme is based on the same physical principles as up-conversion-aided quantum teleportation (Ref. [42, 58]), the authors describe the state to be transferred as that of a single photon A. This photon is one of the many identical photons of the strong coherent field where information is encoded. On page 6, it is mentioned that although the state to be transferred is not destroyed due to the existence of multiple copies, Alice cannot use them. To this purpose, they introduce Charlie as a party that prepares the coherent state, while Alice (the sender) merely "directs photon A and photon B to the non-linear crystal". This suggests that Charlie can select a single photon of the strong laser field and send it to Alice, who somehow can only access this photon and not the rest of the coherent field. Meanwhile, the large number of photons in the coherent state act as an efficiency enhancer in the measurement that Alice performs. This protocol is contradictory and does not describe the experiment, which requires

a different theoretical treatment to the one presented. For example, to establish an appropriate definition of selecting or removing one photon from a coherent field should consider that a coherent state implies the existence of a probability where the photon occupation number is zero. Even more, substituting the single photon field with a coherent state raises important concerns about the security of the protocol and its superiority over classical methods. The introduction of Charlie in the protocol seems like an attempt to conceal the fact that in any practical scenario, an eavesdropper could easily tomograph the coherent state and extract the information to be transmitted. This issue must be addressed in the text, especially considering claims such as "our experiment is based on the same physics, utilizing entanglement and classical communication to securely transmit a quantum state from one system to another" or "it securely transmits the state of the laser photons to the distant and previously entangled photon".

We can appreciate this as a concern as we ourselves had anticipated it. This is why in the first version we had tried to acknowledge this by the detailed discussion that explains how the scheme is similar to but not exactly teleportation, allowing the reader to make up his/her own mind. It is clear from your phrasing of the concern and the queries that follow, that our strategy is confusing to the reader and can be viewed as incorrect if read in a particular way. For example, when we said that Alice could not send any photon, we meant that Bob was not expecting any photon from the channel, hence a cheating sender would not affect the process. Also, we used Charlie only to show that Alice does not need to know the state to be sent but it is evident that she could measure some of the copies and know it.

To address the concern, we have removed all suggestions that this is teleportation, removed all reference to Charlie, and all suggestions that the input prepared state is a single photon. What now follows is the scheme exactly as we did it: Bob sends a single photon from an entangled pair to Alice, who overlaps it with her coherent beam in a nonlinear crystal. The single photon from Bob naturally has no information of what Alice is preparing and is entirely quantum because it requires an entangled pair for Bob to retrieve any information. This allows Bob to retrieve the information using the second photon from his entangled pair. We assume that Alice and Bob are trusted nodes so it is no problem that Alice can know what she is sending, BUT, and this is a crucial point, in our scheme it is not necessary for Alice to know the state to be sent. She can know it but she doesn't have to know it for the scheme to work. This is crucial because the proposed configuration would turn into quantum teleportation directly if the nonlinear efficiencies ever improve so Alice can send the teleported state in a single photon. We think it is clear from our experiment that she is not cheating in any way, so we can conclude that she could know, but doesn't need to. This is why we said that strictly speaking our scheme is neither remote state preparation nor teleportation.

Coming back to the concept as it is now drawn, any eavesdropper (Eve) can only intercept the single photon from Bob which has no information content anyway, so in this sense it is secure, and information is transported from Alice to Bob by a single photon from an entangled pair travelling in the opposite direction with no information. Although Alice uses a coherent state, one photon from this source is used with the one photon from Bob to create the upconverted single photon, this ensured by virtue of conditioning on coincidences with the other photon of Bob's. We now make clear that Alice uses a coherent state and use the term "photon-state A" to describe the photon involved in this upconversion process. Our view is that this is still a

major advance because it is an interesting and novel spatial quantum transport scheme that is both basis and dimension independent, and moves the field of high-dimensional state control forward by introducing a nonlinear detector scheme for spatial modes of light, something that will surely capture the attention of a very large community interested in quantum structured light. Although conceptually the idea is very elegant, it was nevertheless an extremely challenging experiment to execute. As remarked by reviewer #1, this is why we have very extensive SI for both the theory and experiment, for others to quickly follow. We think that you phrased the technical aspect of the advance excellently a little later in your comments: “The experiment demonstrates how non-linear parametric processes allow for arbitrary high-dimensional state projections in the spatial degree of freedom, where a strong coherent field enhances the probability of the process to efficiently characterize the operation and probe its limitations.” Our new revised version highlights the novelty and impact of what we have achieved and we hope you will find it conceptually and factually accurate.

I also find problematic the comparison of the fidelity of the transfer against the classical case, i.e., having no entanglement between photon B and C. The authors consider a classical bound that assumes only one copy of the state (Phys. Lett. A 253, 249-251 (1999), Ref. [52], Phys. Rev. Lett. 72, 797 (1994), and Phys. Rev. Lett. 74, 1259 (1995)), but the protocol uses a coherent state to encode the information (i.e., there are many copies of the state in question). As such, there doesn't seem to be an advantage over the case of Alice using all the available multiple copies to accurately and efficiently tomograph the state and send the result to Bob through a classical channel. The fact that a quantum resource such as entanglement can be used, does not necessarily mean that there is an advantage over the classical counterpart.

We hope the answer to the first part of the question now answers this too. The resource without any Charlie involved is an entangled pair of photons, one kept in Bob's lab for state measurement and the other sent to Alice who wishes to transport a high-dimensional state to Bob. Our calculations are then on the channel capacity of this link, entirely quantum, and the quantum state as measured by Bob. If Alice were to directly send her information classically then of course this would not be quantum transport and Eve could easily intercept and copy the state. We have introduced a new concept figure, new Fig. 1, that hopefully makes this very clear.

With all of this, I cannot recommend publishing the paper in its current state because it presents the work as a “versatile and scalable” alternative to exchanging high-dimensional information through quantum teleportation. This comparison is inaccurate because the two schemes have conceptual differences, they don't provide the same communication advantages and have very different relevance in the field of quantum technologies. The implication of introducing quantum transport as a strategy that “converges to high-dimensional quantum teleportation in the limit of a single photon input” misleads the reader on its relevance and application, undermining the significance of any previous or future demonstrations of high-dimensional quantum state teleportation. The authors should reformulate the theoretical framework in the "Experimental concept" section and Supplementary Notes II and X, addressing the problems I've pointed out and being clear about the use of a coherent field in the experimental setup in Fig. 1.a. They also need to indicate the existence and accessibility of the many copies of the state, and how they invalidate the

transport properties mentioned in the last paragraph on page 5. Furthermore, they must change the narrative in the section "From transport to teleport" and remove direct comparisons to the quantum teleportation results of Ref. [45-46] on page 4 (they incorrectly claim to have replicated the state-of-the-art results in both two and three dimensions), and in Supplementary Note XI (benchmarking the transport protocol by comparing it to high-dimensional teleportation demonstrations does not make sense).

We agree. We have removed and/or reformulated (each as per your request) all the material you mentioned above and indeed completely agree that it is unfair to say we have replicated the prior art as these two studies were very different and truly used single photons from Charlie, and removed all comparison. We think you will be happy with the new version in its toned down form. As both you and review #1 said, present technology does not allow two single photon inputs to the upconversion crystal due to efficiency issues, so one of the photon "ports" has to be a coherent state, and this can only be the port with the information to be sent. This means it is not teleportation. But since Alice need not know the state for the process to work, we would like to alert the reader to the fact that it could be in the future – who can say what will be possible? For example, using a nonlinear process with three inputs and not just the two of SFG might allow the third "port" to be a coherent seed beam that only drives efficiency (a plane wave with no information in its structure) while the two other inputs could then be single photons. Or maybe indeed metasurfaces reach many orders of magnitude efficiency gains. We just don't know. To this end, we have moved a modified version of the section "From transport to teleport" from the main text to the SI for any interested reader who wishes to understand what would have to be done for this to reach teleportation – we think this is fair and also interesting (see your comment later). Please find these amends highlighted in the red-lined version of the revised manuscript, and here is a list of **sentences we have removed** to avoid misleading the reader:

- "The general scheme for teleportation is used here ..."
- "However, despite this technological limitation, our experiment is built on the same physics, leveraging entanglement and classical communication to securely transfer a quantum state from one system to another."
- "... and despite encoding the quantum state in many copies, our scheme requires the same ingredients of the quantum teleportation protocols to transfer such spatial information."
- "*To benchmark our protocol with respect to other reported high-dimensional quantum transport protocols*, a state tomography on each of the 12 MUB states for *the current* three-dimensional *limit* was performed."
- "Corresponding results for the 12 MUB states reported in Refs. [PRL 123, 070505 (2019), PRL 125, 230501 (2020)] are shown in columns to the right for comparison and a final fidelity, averaged over all the states, given at the bottom. It follows, that our protocol compares well with those reported where $F_{ave} = 0.895 \pm 0.042$ in our scheme compares to $F_{ave} \approx 0.75 \pm 0.055$ [PRL 123, 070505 (2019)] and $F_{ave} = 0.70 \pm 0.026$ [PRL 125, 230501 (2020)]."
- "Both linear teleportation and our nonlinear quantum transport scheme require an entangled photon pair as a quantum resource, but linear optical approaches for qudit teleportation require ancillary photons in a manner that the experimental configuration is hard-coded for a particular channel's dimensionality."

After executing these changes, the study could still be of relevance to the field. The experiment demonstrates how non-linear parametric processes allow for arbitrary high-dimensional state projections in the spatial degree of freedom, where a strong coherent field enhances the probability of the process to efficiently characterize the operation and probe its limitations. The technique could be put in context when comparing it with measurement strategies in the time-frequency domain that use SFG for the selection of arbitrary temporal modes (See the Quantum Pulse Gate [Opt. Express 19, 13770 (2011), Phys. Rev. A 90, 030302 (2014)], or the coherent projections on time-bin encoded photons [PRL 111, 153602 (2013)]).

We would like to thank the reviewer for the very balanced report on our work, pointing out issues but also here pointing out the novelty and importance of the advance. We completely agree and rather than a lengthy discussion on teleportation we now have replaced it with a discussion on where nonlinear quantum optics might go in the future, and how this could fuel advances in quantum structured states from space to time degrees of freedom. You will find the abstract, introduction and discussion all reflect this change.

Furthermore, studying the spatial mode capabilities of the up-conversion channel could prove useful when advancements in up-conversion efficiencies permit real demonstrations of high-dimensional quantum teleportation through non-linear schemes.

Thank you for this comment - we agree. For this reason, we have kept the “from transport to teleport” but in a manner that does not compare it to the prior art, and have moved it to the SI so as not to mix the central message of the main text.

However, some important technical aspects require attention:

1) For the two-dimensional state, coherent transport is shown with the introduction of a modal phase angle between photons A and B and the calculation of fringe visibilities. On the other side, three-dimensional transport is confirmed with a fidelity estimation through state tomography. Since a relationship between the visibilities and the state fidelity is not provided, a comparison of the transfer in these two different dimensions can't be made. The authors should then calculate the fidelity for the two-dimensional case.

Thank you, we have now calculated the fidelity related to the two-dimensional fringe visibility and included this information in the main text as follows: The resulting visibilities (V) allow us to determine the fidelity [39] $F = (1+V)/2$, with raw values varying from 90% to 93%, and background subtracted all above 98%.

2) Tomographic measurements support three-dimensional transfer. However, results in dimensions greater than three rely on a metric called similarity. This figure of merit is not standard in quantum state characterisation, and it doesn't demonstrate a coherent transfer of the state because it lacks phase information, a crucial aspect of quantum communication schemes. The authors need to calculate the state fidelity in the four and nine-dimensional cases, either with complete tomographic measurements or with the use of a witness. Otherwise, any claim of quantum transport beyond dimension three should be removed.

Thank you for this. First, a full tomography of high-dimensional states is prohibitively time consuming in even standard two-photon experiments, but here the very low counts meant that it was impossible to do so. As mentioned by reviewer #1, this was the first such experimental demonstration of this approach so surely we and others will refine it in future versions of the experiment. Instead we characterised the channel capacity using a fast measure of entanglement dimensionality (old Ref [51], now [36]), so we can say that we have a channel that supports up to 15 dimensions. This measure itself includes superpositions with phases so confirms a coherent process up to $d = 15$. Second, you are correct that similarity is not a usual term to be used. We agree that the phase information is not captured in the similarity, even though we can assume it is there by virtue of the channel capacity measure and the nature of the experiment. Note that our results for $d = 2, 3$ and 4 do include phase information in preparation and measurement, and thus we can consider these examples of coherent transfer of the state, with the core data in the old Fig. 3 (now Fig. 4) and the SI. That leaves only the $d = 9$ case in the last two figures. Because $d = 9$ is within the $d = 15$ channel capacity it can be considered fully coherent too. For example, below we extract the $d = 9$ data from the channel capacity measurements, and the last point in the bottom right indicates the inferred dimension of this state.

In each subfigure, the four curves represent measurement settings for Alice $\theta_A = (0, \pi/2, \pi, 3\pi/2)$ while θ_B is the measurement setting for Bob, controlling the relative phases, while the projections are coherent superpositions with phases of up to 9 modes. Note that the modal spacing in the superpositions used for these particular measurements are 0 (top left), 2 (top right), and 4 (bottom left), meaning that the non-zero modes contributing to the

projection will be multiples of 1, 3 and 5, respectively (see Methods and [36]). This is why we are sure we have coherent quantum transport even in $d = 9$.

BUT, since we did not perform phase transport as a prepared and measured state beyond $d = 4$ we have reduced the claim in the main text and caption to be on the safe and conservative side. Because it is not possible now to return to the experiment any time soon (the student and post-doc have moved on and we start from fresh) we will follow your advice and claim a *quantum transport scheme with a measured channel capacity of $d = 15$, demonstrated by coherent transfer up to four dimensions*. We have added a comment on similarity and the reason for using it, but with the acknowledgement that it doesn't account for phase.

3) The results in Fig. 3.c) and Fig. 5 only show diagonal terms of the crosstalk matrix, which convey the invalid notion that they are sufficient to confirm the "agreement" between encoded spatial information and the projected state. These results could also describe the transfer of a maximally mixed state. The authors should include full crosstalk measurements between prepared and measured states to support their claims of quantum transport.

Old Fig. 3(c) (now Fig. 4(c)) does include the full crosstalk matrix. It is just that the diagonal terms dominate. We have now changed this plot also following the question of reviewer #1 to make the $d = 4$ case clearer. Yes, we agree concerning the old Fig. 5 (now Fig. 6). The results there only confirm that the diagonal elements of the density matrices of the input states were transferred correctly. Consequently, they cannot evidence agreement between the coherences (off-diagonal elements of the density matrices) before and after quantum transport. This is now pointed out in the capture of this figure. We point out again that the measure of the channel capacity itself confirms that the process is coherent up to $d = 15$ by virtue of how this measure works (see example plots for $d = 9$ above).

4) In Figs. 3-5, transparent bars are said to describe the prepared states. However, what do coincidence counts of the prepared state even mean?

Thank you, we have now added text at the end of the Results section to explain that this refers to the normalised probability, which can be expressed as counts for visualisation.

Additionally, please have a careful read of the manuscript and correct the typos. For example:

- In the abstract: scaleable.
- In the caption for Fig. 1: "respectively. and The cross-talk plots"
- On page 6, last paragraph: securily
- Supplementary Note IX, Quality of the entanglement channel: "projected onto in the the sender arm"

Thank you, these are now all fixed and the manuscript carefully proofread.

As a concluding remark, we are excited to see this advance published so have made a concerted effort to address all the important issues you have raised. We look forward to hearing if this makes for a more fair and compelling story.

REVIEWERS' COMMENTS

Reviewer #1 (Remarks to the Author):

I thank the authors for taking my comments as well as the comments of referee 2 serious and adjusting the manuscript accordingly. While I was not as critical as referee 2 in terms of possible confusion with actual high-dimensional teleportation, I agree to some extent with the other referee that in the earlier form it could have been misinterpreted as equally powerful as actual high-dimensional teleportation (although the authors did not explicitly say it nor did I interpret it like that) thereby lowering the significance of earlier and possible future demonstrations of high-dimensional teleportation. In its revised form, however, I think the differences are clear such that I think that the implemented changes put the manuscript in a better context thereby making the experiment as well as the results easier to understand. Hence, I reiterate my earlier suggestion of publication of the manuscript as I think it is an interesting scheme which might inspire novel studies combining structured light and nonlinear optics in the quantum domain.

There are two smaller comments, which I hope the authors can address:

- In different figures (fig4-6), the authors compare their measurements in coincidence counts with theoretically expected probabilities. After a remark by the other referee, the authors now explicitly mention it in the text (page 5), which is nice, however, I don't see how the normalization was done? It looks like the authors always assume one of the measurements to be correct, i.e. as a kind of reference, and then adjust the transparent bars for the others. How did the authors decide, which one is the reference? This needs to be clarified and explained in the text.

- In supplementary section XV, the authors still have a third-party "Charlie" to describe the differences between their transport and actual teleportation. This is fine, however, since there is no Charlie introduced in the main text anymore (which is good), I think a sentence introducing the idea of a third party called Charlie would be good. In its current form Charlie appears a bit out of nowhere.

Reviewer #2 (Remarks to the Author):

After reading the revised version of the manuscript and the response to my comments, I feel the implemented changes now present a fair and compelling story that highlights the relevance of the work without inaccurate comparisons or misleading claims. In addition, the technical concerns have been addressed to fairly present the results, with clarity on the capabilities of the technique and the limitations of its characterisation.

I therefore recommend the publication of the manuscript.

RESPONSE TO THE REVIEWER COMMENTS

Reviewer #1 (Remarks to the Author):

I thank the authors for taking my comments as well as the comments of referee 2 serious and adjusting the manuscript accordingly. While I was not as critical as referee 2 in terms of possible confusion with actual high-dimensional teleportation, I agree to some extent with the other referee that in the earlier form it could have been misinterpreted as equally powerful as actual high-dimensional teleportation (although the authors did not explicitly say it nor did I interpret it like that) thereby lowering the significance of earlier and possible future demonstrations of high-dimensional teleportation. In its revised form, however, I think the differences are clear such that I think that the implemented changes put the manuscript in a better context thereby making the experiment as well as the results easier to understand. Hence, I reiterate my earlier suggestion of publication of the manuscript as I think it is an interesting scheme which might inspire novel studies combining structured light and nonlinear optics in the quantum domain.

There is two smaller comments, which I hope the authors can address:

- In different figures (fig4-6), the authors compare their measurements in coincidence counts with theoretically expected probabilities. After a remark by the other referee, the authors now explicitly mention it in the text (page 5), which is nice, however, I don't see how the normalization was done? It looks like the authors always assume one of the measurements to be correct, i.e. as a kind of reference, and then adjust the transparent bars for the others. How did the authors decide, which one is the reference? This needs to be clarified and explained in the text.

We would like to thank the reviewer again, this time also for the careful read of the reviewer #2 comments, our reply and revised version of the manuscript. We have now added a sentence below Fig. 4 in the main text, describing how and why the normalization of the prepared states was performed: *"Note that the prepared states (transparent bars) in the figures throughout the letter are obtained by the averaged sum of all measured values involved, facilitating comparison with the raw coincidences."*

- In supplementary section XV, the authors still have a third-party "Charlie" to describe the differences between their transport and actual teleportation. This is fine, however, since there is no Charlie introduced in the main text anymore (which is good), I think a sentence introducing the idea of a third party called Charlie would be good. In its current form Charlie appears a bit out of nowhere.

Thank you very much for bringing this to our attention. We have added an explanation in the Supplementary Note 15 now, describing why a third-party encoding the state has been considered in this particular example: *"We consider in this example a third-party, i.e., Charlie, preparing the state to be transferred, so to emphasize in the similarity between our quantum transport configuration and the quantum teleportation, where such high-dimensional spatial state could be encoded in a single photon (see Suppl. Fig. 27 (b)). In any of the cases Alice does not need to know the state that is sent to her, never encodes information on any photons, and in our particular case never even sends any photons to Bob."*

Reviewer #2 (Remarks to the Author):

After reading the revised version of the manuscript and the response to my comments, I feel the implemented changes now present a fair and compelling story that highlights the relevance of the work without inaccurate comparisons or misleading claims. In addition, the technical concerns have been addressed to fairly present the results, with clarity on the capabilities of the technique and the limitations of its characterisation.

I therefore recommend the publication of the manuscript.

Thank you very much for the positive assessment of our revised manuscript and the recommendation for publication. We appreciated the previous comments, as they helped to improve the message placing our work in a better position with respect other related studies in the field. We do hope these results will encourage new advances in the field of nonlinear detection, pushing the limits of the quantum transport of high-dimensional states.